# Comparative Study between Two Simple Synthesis Methods for Obtaining Green Gold Nanoparticles Decorating Silica Particles with Antibacterial Activity

**DOI:** 10.3390/ma15217635

**Published:** 2022-10-30

**Authors:** Karen M. Soto, Angelica Gódinez-Oviedo, José. M. López-Romero, Eric. M. Rivera-Muñoz, Edgar Jose López-Naranjo, Sandra Mendoza-Díaz, Alejandro Manzano-Ramírez

**Affiliations:** 1Centro de Investigaciones y de Estudios Avanzados del I.P.N., Unidad Querétaro, Querétaro 76230, Mexico; 2Departamento de Investigación y Posgrado en Alimentos, Facultad de Química, Universidad Autónoma de Querétaro, Querétaro 76010, Mexico; 3Centro de Física Aplicada y Tecnología Avanzada, Universidad Nacional Autónoma de México, Querétaro 76000, Mexico; 4Departamento de Ingeniería de Proyectos-CUCEI, Universidad de Guadalajara, Guadalajara 44100, Mexico

**Keywords:** green synthesis, AuNPs, silica particles, antibacterial activity

## Abstract

The SiO_2_ particles system is one of the most common ways to protect colloidal metal systems, such as gold nanoparticles, from aggregation and activity loss due to their high chemical stability and low reactivity. In this study, silica green gold nanoparticles (AuNPs synthesized with mullein extract) were fabricated using two different sol–gel methods. The nanoparticles were characterized by Scanning Electron Microscopy (SEM), X-ray diffraction (XRD), Fourier Transformed Infrared (FTIR), and the antibacterial activity against pathogens (*Staphylococcus aureus*, *Listeria monocytogenes*, *Escherichia coli*, and *Salmonella enterica*). Synthesis-1 nanoparticles had a kidney-shaped form and uniform distribution, while synthesis-2 nanoparticles had a spherical and non-uniform form. Characterization showed that temperature is an important factor in the distribution of AuNPs in silica; a decrease allowed the formation of Janus-type, and an increase showed a higher concentration of gold in energy-dispersive spectroscopy (EDS) analysis. Overall, similar bands of the two synthesis silica nanoparticles were observed in FTIR, while XRD spectra showed differences in the preferential growth in AuNPs depending on the synthesis. Higher antibacterial activity was observed against *S. aureus*, which was followed by *L. monocytogenes*. No differences were observed in the antibacterial activity between the two different sol–gel methods.

## 1. Introduction

Gold nanoparticles are nanomaterials that have attracted significant attention in recent years due to their unique characteristics, such as large surface area, low toxicity, easy synthesis, functionalization, and optical, physical, anticarcinogenic, antibacterial, and electrochemical properties, among others. They have multiple biotechnology applications for imaging, biosensing, and gene and drug delivery [1,2]. In recent years, the synthesis of AuNPs must be accompanied by eco-friendly, cheap, and novel approaches to minimize or altogether avoid the administration of dangerous chemicals and simultaneously diminish the accumulation of hazardous wastes. Safer production alternatives include applying biological materials such as plant extracts or biomolecules of plants, bacteria, fungi, or their lysates [3,4].

Despite the multiple applications, its use in biomedical applications has different limitations, such as its low stability in organic fluids (blood, urine, among others), the surface restriction for binding biomolecules (proteins), and some cytotoxicity. Yue et al. (2019) studied the effects of NP size, NP concentration, temperature, and surface modification in the aggregation of AuNPs in blood and distilled water, observing a significant aggregation in blood with a temperature increase [5]. The encapsulation of AuNPs with a protecting shield to stabilize nanoparticles and develop, for instance, at the same time, some surface functionalization, leaching, facilitate handling, improves mechanical and thermal properties, which increases their applications [6]. Protecting AuNPs from coagulation can be completed by core–shell techniques, for example, by a polymer coating usually made of PVP, PEG, or silica particles. 

Colloidal silica particles present different properties, including high hydrophilicity, high mechanical stability, and high biocompatibility, which makes them excellent in biomedical and industrial applications, used for photoacoustic imaging, cell tracking, targeted drug delivery, catalyst, and photonics [7] The sol–gel process is the most common methodology to produce pure silica nanoparticles, due to the facile synthesis and tunable properties, through systematic monitoring of reaction parameters. This method uses quaternary alkylammonium surfactants or triblock copolymers as templates in a mixture of alcohol, water, and ammonia [8]. The basic strategy is based on the supramolecular self-assembly of the templates at higher concentrations than critical micellar concentrations (CMC) and in alkaline or acidic conditions, hydrolyzed inorganic silica precursors such as tetraethylorthosilicate (TEOS) [9,10,11,12]. The size and morphology are essential for its possible applications; these characteristics can be modified by temperature, pH, and concentration changes. Previous reports of gold decorating silica nanoparticles developed are reported for Surface-Enhanced Raman Scattering (SERS) applications [13], plasmonic properties [14], and cancer detection [15].

In the present work, two synthesis methods were compared to produce silica nanoparticles on which gold nanoparticles made by green synthesis with mullein extracts were deposited to improve some of their properties, protect them from aggregation and maintain their properties. The systems were physically and chemically characterized, and the stability of AuNPs incorporated in silica nanoparticles and the colloidal solution was studied for six weeks. Their antibacterial activity against Gram-positive and negative microorganisms of sanitary and food importance was evaluated. This work is the first report that in situ plant green synthesis AuNPs decorating Janus nanoparticles for antibacterial activity, which are results that may be of significance for biomedical and agri-food applications. 

## 2. Materials and Methods

### 2.1. Materials 

Tetraethyl orthosilicate (TEOS) (99%), sodium borohydride (NaBH_4_, 98%), gold (III) chloride hydrate (HAuCl_4_), cetyltrimethylammonium bromide (CTAB), and absolute ethanol were obtained from Sigma Chemical Co. (St. Louis, MO, USA). Muller Hilton agar from Bioxon (Cuautitlán Izcalli, Mex, Mex). Tryptic soy broth was produced from Becton, Dickinson & Company (Franklin Lakes, NJ, USA).

### 2.2. Extract Preparation 

Common mullein (*Verbascum thapsus*) (World Checklist of Selected Plant Families, WCSP record in review: Sp. Pl. 177 1753) flowers were collected in Queretaro (Mexico) and deposited in the “Dr. Jerzy Rzedowski” (QMEX) herbarium of Universidad Autónoma de Querétaro (Juriquilla, Querétaro, Mexico). Mullein flowers were washed with deionized water and dried in a convection oven at 60 °C for 6 h. The dried residues were then powdered to fine particles using a mixer grinder (KRUPS GX4100) and passed through a 60-mesh screen (250 µm particle size). Ten grams of the powdered mullein flowers were added to 100 mL ethanol and left to stand for 4 h. The extracts were filtered (Whatman Paper no. 40) and stored at 4 °C for further analysis [16].

### 2.3. Synthesis-1 of Silica-Gold Nanoparticles

Silica nanoparticles were synthesized using a modified sol–gel process described by Goncalves et al. (2019) with some modifications. In a polypropylene flask were added deionized water (240 mL) and NaOH solution (1.7 M, 1.75 mL); the mixture was stirred until the temperature reached 32 °C. Afterward, CTAB (0.5 g) was added and stirred for 30 min; later, a TEOS (2.5 mL) was added dropwise, and the solution was left stirring for three hours. The dispersion obtained after synthesis was filtered with a vacuum filtration system using a Whatman Paper no. 40, and the solid was washed with ethanol and water (50% *v*/*v*). The solid obtained was dried, obtaining the blank S1 [11]. In the case of silica gold nanoparticles, 15 mL of deionized water was replaced by HAuCl_4_ (0.5 mM), and 50 µL of mullein extract was added for the gold nanoparticle synthesis to obtain Mullein S1.

### 2.4. Synthesis-2 of Silica-Gold Nanoparticles

For synthesis-2, we followed the methodology described by Vega et al. (2019) with some modifications. In a round-bottomed flask connected to a reflux condenser, we added 6 mL of HAuCl_4_ (0.5 mM) and CTAB (11.6 mg), and the temperature was raised to 80 °C; later, 43 µL of NaOH (2 M), and 20 µL of mullein extract were added. The solution was stirred for a few minutes before the dropwise addition of 60 µL of TEOS. The mixture was stirred at 80 °C for 2 h. After this time, the mixture was allowed to cool to room temperature, and the silica was filtered or centrifuged (10 min at 3000 rpm) and washed with water several times. In the case of blank nanoparticles, the HAuCl_4_ was replaced with water, and no extract was added [17]. 

### 2.5. Characterization 

#### 2.5.1. Size and Morphology

The morphology and diameters of silica and silica gold particles were determined using a scanning transmission electron microscope (STEM SU8230, Hitachi, Tokyo, Japan) and a scanning electron microscope (SEM JXA—8530F, JEOL, Tokyo, Japan). The diameters of the AuNPs were obtained by measuring at least 50 particles from STEM images with Image J software bundled with Java 8 (Stapleton, NY, USA). 

#### 2.5.2. Fourier Transform Infrared Spectroscopy (FT-IR)

FT-IR spectroscopy (Spectrometer Spectrum Two, PerkinElmer, Waltham, MA, USA). was performed to characterize the present and change of the functional groups of the silica gold nanoparticles after modification. The particles were mixed with KBr powder and then pressed into a thin slice for testing. The spectra were collected in a wavelength range of 4000–400 cm^−1^ by averaging 32 scans at a resolution of 4 cm^−1^.

#### 2.5.3. X-ray Diffraction (XRD)

The XRD (Ultima IV, Rigaku, Tokyo, Japan) of the silica and silica gold nanoparticles was performed at room temperature in reflection mode using Cu-Kα (k = 1.5418 °A) radiation at 40 kV and 40 mA. The 2θ scan data were collected at a rate of 2°/min within the scattering range of 10–50°.

#### 2.5.4. UV-Vis Analysis 

For UV-Vis analysis, the silica powder was suspended and sonicated in deionized water, and the UV-Vis spectra were obtained in a Spectra Max Tunable Microplate Reader (Molecular Devices Co., Sunnyvale, CA, USA). 

#### 2.5.5. Stability 

The colloidal solutions obtained after biosynthesis were stored at room temperature for 6 weeks without light; UV-Visible spectra (Spectra Max Tunable Microplate Reader, Molecular Devices Co., Sunnyvale, CA, USA) were recorded at different times. For AuNPs incorporated in silica structures, the silica powder was suspended and sonicated in deionized water stored at room temperature for 6 weeks without light; the UV-Vis spectra were obtained. 

#### 2.5.6. Antibacterial Activity 

To evaluate the antibacterial activity of the silica–gold nanoparticles, two Gram-negative bacteria: *Escherichia coli* ATCC 11229 and *Salmonella enterica* serotype Typhimurium ATCC 14028, and two Gram-positive bacteria: *Listeria monocytogenes* ATCC 19115 and *Staphylococcus aureus* ATCC 6538 strains (obtained from the LECRIMA laboratory, Autonomous University of Queretaro, Mexico), were cultivated in tryptic soy broth and incubated at 35 °C/24 h. The cell suspension was centrifuged (3500 rpm, 10 min), and the resulting pellet was washed twice in 3 mL of saline solution (0.85% NaCl). The suspension was adjusted to the 0.5 McFarland standard. 

The antibacterial susceptibility was performed using the disk diffusion technique on Muller Hilton agar (Bioxon, Cuautitlán Izcalli, Mexico), following the methodology of the Clinical and Laboratory Standards Institute [18]. The silica–gold nanoparticles were suspended in Milli Q water, and five µL of each treatment were put on the agar surface. The plates were incubated for 24 h at 37 °C, and the zone of inhibition was measured. 

## 3. Results and Discussion

### 3.1. Size and Morphology 

SEM images in Figure 1 show the morphologies of silica and silica–gold nanoparticles. In synthesis-1 (Figure 1a,c), kidney-shaped particles with diameters around 196 nm were obtained, while in synthesis-2, the formation of spherical and some elongated particles is observed, with larger diameters than S1, with a mean diameter of 560.32 nm and a non-uniform distribution. The incorporation of the gold particles on the silica substrate was carried out by performing the synthesis of both particles at the same time; in synthesis-1, we can see that when particles are formed with mullein extract, there is an apparent decrease in particle size. In the case of synthesis-2 (Figure 1b,d), the mean diameter decreases more when incorporating the synthesis of AuNPs. 

STEM images in Figure 2 show the distribution of the gold particles in the silica substrate. Essential differences can be observed in both methods, since in synthesis-1 (Figure 2a,c), the gold particles will be found only at one end of the silica particles, forming particles called Janus. In contrast, in S2 (Figure 2b,c), we observe a more significant visible amount of gold particles on the silica substrates. These do not have a homogeneous distribution. In addition, due to the aggregates formed, we cannot differentiate if the particles of gold are either embedded or on the surface of the silica. The main differences between both syntheses are the reaction temperature and the amount of hydroxide which modifies the pH of the reaction. These are essential factors in the hydrolysis of TEOS, which increases at high temperatures. As a result, particle size decrease, but the increased rate of hydrolysis reaction sometimes results in larger particles when the concentration of the basic solution increases [7,8,12], similar to what was observed in synthesis-2.

The size of the gold nanoparticles differed depending on the type of synthesis; in one, there was an average diameter of 7.851 ± 2.36 nm, while in 2, the diameter increased to 11.49 ± 4.41 nm (Table 1). These differences are related to changes in temperature, which increased 50 °C in synthesis-2; this increment in temperature increases the particles’ nucleation, leading to larger aggregates forming [19].

### 3.2. X-ray Diffraction (XRD) 

The XRD of silica–gold nanoparticles is shown in Figure 3. The broad band at 2θ between 15° and 30° is associated with the amorphous SiO_2_ corresponding to silica particles. The reflection peaks appeared at 38.64, 44.34, 64.9, and 77.36 corresponding to (1 1 1), (2 0 0), (2 2 0), and (3 1 1) Miller indices, respectively; these lattice planes correspond to the standard face-centered cubic (FCC) phase of metallic gold nanostructures, further indicating the formation of crystalline AuNPs. We can observe an essential difference between the two synthesis methods; i.e., in synthesis-1, they grow preferentially in the (200) direction. As may be observed, there is anisotropic growth and other shapes, such as nanorods, nanocubes, and nanoprisms. In synthesis-2, the gold particles grow in the (111) direction related to a constant growth, where the particle size is favored, and spheroid particles are obtained. The heterogeneous nucleation of the gold nanoparticles occurs on the silica particles, and various factors can increase the nucleation rate. One of them is the temperature; in some cases, the nucleation rates tend to decrease with temperature and affect the morphology and size [13,20].

### 3.3. Fourier Transform Infrared Spectroscopy (FTIR)

FTIR analyses of silica particles and silica–gold particles of synthesis-1 and 2 are shown in Figure 4. We can observe characteristic absorption bands in 3000–2975 cm^−1^ corresponding to the stretching vibration of Si-OH; 1616 cm^−1^ related to molecular water physically adsorbed. While 1414 cm^−1^ corresponds to the symmetric deformation vibration of C-H; 1084 cm^−1^ asymmetric stretching vibration Si-O-Si; 954 cm^−1^ is associated with the stretching vibration of Si-OH; 779 cm^−1^ corresponds to symmetric stretching vibrations Si-O-Si; 568 cm^−1^ related to stretching vibration of Si-O. In the spectrum of the silica–gold particles synthesized with two methods, the intensity of Si-O-Si (1084 and 779 cm^−1^) and Si-OH (949 cm^−1^) peaks have been reduced to smaller wavenumbers significantly, indicating the presence of gold shells in silica particles [21]. Bands related to plant extracts may not be visible due to low concentrations of AuNPs. 

### 3.4. Stability of AuNPs

The UV-Vis spectra of AuNPs synthesized with mullein extract are shown in Figure 5a, while the stability of AuNPs incorporated in silica particles by synthesis-1 are shown in Figure 5b and those by synthesis-2 are shown in Figure 5c. We can observe that the AuNPs incorporated in Janus silica nanoparticles obtained in synthesis-1 present better stability after storage per 6 weeks, which is followed by AuNPs incorporated in synthesis-2. The colloidal AuNPs without the incorporation in silica structures presented the least stability, since the surface plasmon band is no longer observed after the first week of storage, which is a fact related to the formation of aggregates and loss of stability. Similar results are observed with AuNPs incorporated in silica by synthesis-2 since, from the first week of storage, the surface plasmon band undergoes a widening, which is related to the increase in size and changes in the morphology of the particles. This is less noticeable than in the colloidal AuNPs, losing the total stability in week 6, while in synthesis-1, only changes in absorbance intensity are observed. However, no essential changes in the surface plasmon band can be considered the most stable, and the system confers greater stability, avoiding agglomeration of the particles.

### 3.5. Antibacterial Activity

The antibacterial activity of the silica and silica–gold nanoparticles is shown in Table 2. At first glance, we can see that the highest antibacterial activity was obtained against *S. aureus*, followed by *E. coli*, *L. monocytogenes*, and *S. enterica*. In general, the silica–gold particles presented activity against both Gram-positive and negative bacteria types. However, the major antibacterial activity was observed in the Gram-positive type. Previous studies showed that this type of bacteria tends to accumulate many particles in its cell wall, which could be related to a higher antibacterial activity because the silica particles are larger; they cannot penetrate the cell. However, they can accumulate in the wall, causing some pores. In addition, some gold particles on the surface of the silica particles could be released and go through the wall [22]. In general, no significant differences were observed in both synthesis methods. Although in the STEM images and EDS results, it is observed that there is a more substantial amount of gold particles in synthesis-2. This does not increase the antibacterial activity, which is ascribed to the fact that particles are embedded in the silica, preventing their interaction with bacteria, contrary to synthesis-1, in which the particles are on the surface, allowing them to act against microorganisms. In synthesis-1, particles called Janus are formed, which bear the name of the Roman god and have two compartments/surfaces with different physicochemical properties in a single particle, providing them with better surface characteristics compared to homogeneous particles [23]. Moreover, it is essential to mention that the diameter of the gold particles was different in each method. Being the smallest in synthesis-1 can be released to the media and act as antimicrobial material, as observed by Wu et al. (201) in silica-coated gold−silver nanocages [24]. Different studies report that particles smaller than 10 nm present greater ease of passing through the cell wall of microorganisms that cause significant damage to cell metabolism through the production of reactive oxygen species (ROS) [4,25].

Extracts of the *V. thapsus* (common mullein) have been reported for antifungal, antioxidant, antibacterial, and anti-adherent activities related to the presence of some phenolic compounds, flavonols, and saponins, and present activity against Gram-positive and negative bacteria such as *Klebsiella pneumoniae, Escherichia coli, Pseudomonas aeruginosa*, *Staphylococcus epidermidis, Staphylococcus aureus* and *Streptococcus pyogenes* [26,27].

Figure 6 shows the antibacterial activity corresponding to different treatment concentrations; the action depends on the concentration of the particles. We can observe that the activity is less against Gram-negative bacteria since, from the second concentration, no antibacterial activity is observed in *S. enterica*. We must highlight that the target particles (without AuNPs) of both synthesis present activity against Gram-positive bacteria. This could be related to an accumulation in the cell wall that affects their metabolism. Finally, the activity of the AuNPs without silica was also measured, and a more significant activity against all microorganisms was observed; however, when it was incorporated into the silica particles, its activity did not decrease significantly.

## 4. Conclusions

Two methods of sol–gel synthesis were evaluated for the development of silica nanoparticles and AuNPs, obtaining essential differences in terms of morphology and the position and quantity of gold nanoparticles. In synthesis-1, more homogeneous particles were obtained with bean-shaped AuNPs on one of its sides. In terms of activity, no differences were observed between the methods except for considering that in synthesis-1, the use of temperature is not necessary compared to synthesis-2 (80 °C). The former is considered a more economical, ecological, and easy method for encapsulating AuNPs. More dispersed particles with little aggregation have been obtained, increasing their possible application in different areas. Incorporating the AuNPs in silica did not significantly decrease their antibacterial activity against pathogenic microorganisms that are of sanitary and food importance without providing resistance, allowing their use in biomedical and food applications. The stability of AuNPs synthesized with mullein extract was improved with the incorporation of silica Janus nanoparticles.

## Figures and Tables

**Figure 1 materials-15-07635-f001:**
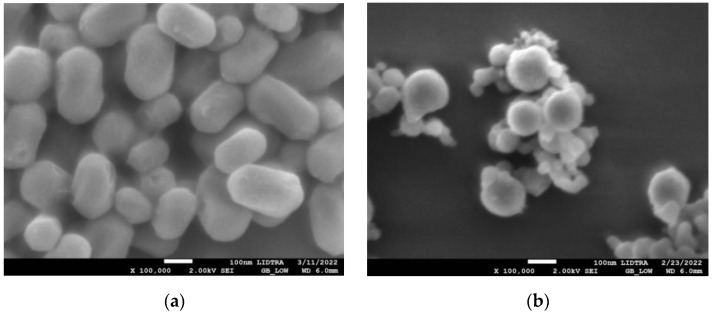
SEM micrographs of silica-gold nanoparticles with two different syntheses (**a**) blank S1, (**b**) blank S2, (**c**) Mullein S1, (**d**) Mullein S2.

**Figure 2 materials-15-07635-f002:**
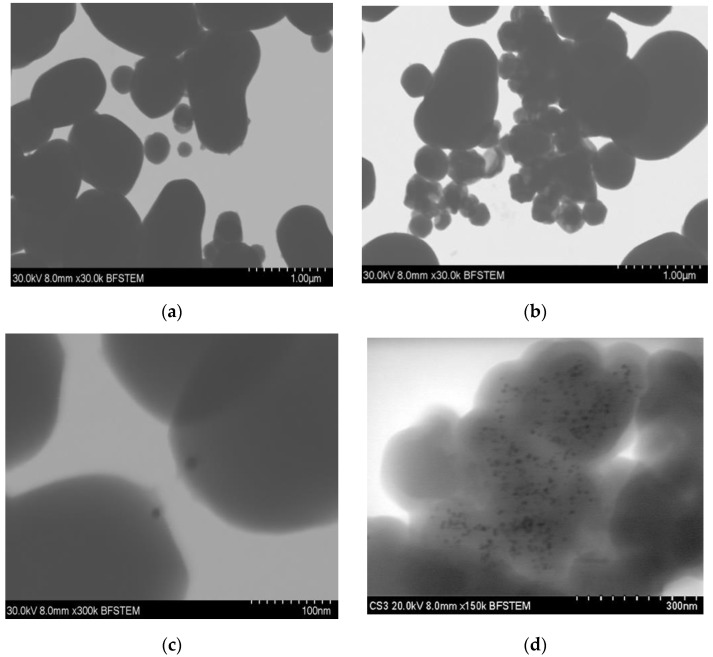
STEM micrographs of silica–gold nanoparticles (**a**) blank S1, (**b**) blank S2, (**c**) Mullein S1, (**d**) Mullein S2.

**Figure 3 materials-15-07635-f003:**
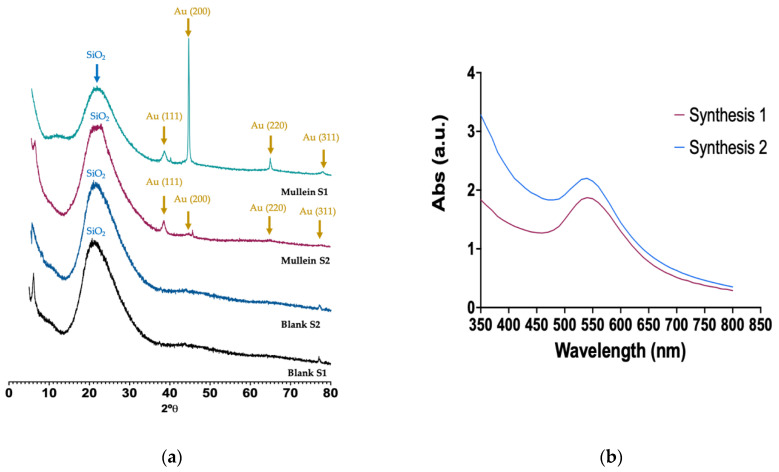
(**a**) X-ray diffraction of silica and silica–gold nanoparticles synthesized with two methods, (**b**) UV-Vis spectra of silica–gold nanoparticles.

**Figure 4 materials-15-07635-f004:**
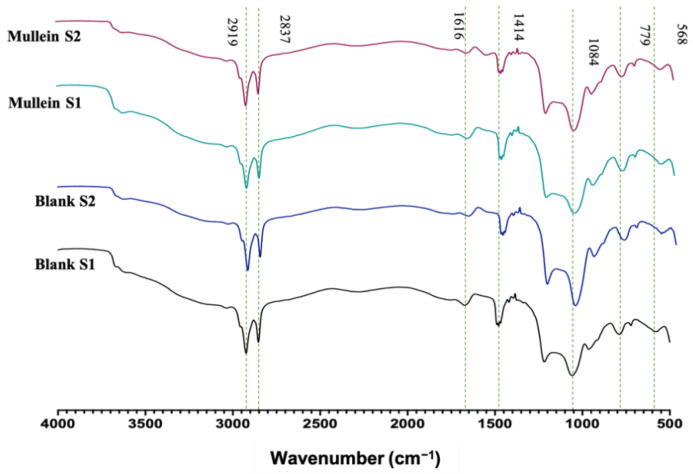
FTIR of silica–gold nanoparticles.

**Figure 5 materials-15-07635-f005:**
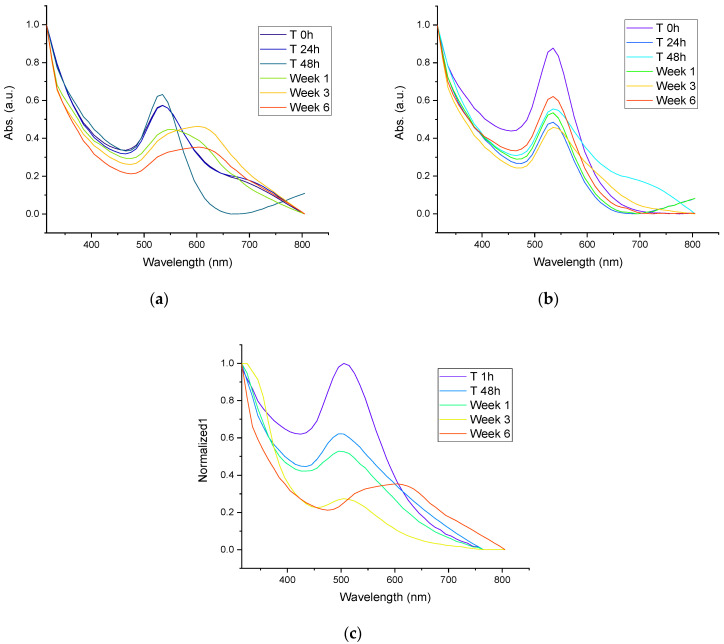
(**a**) UV-Vis spectra of stability of colloidal AuNPs synthesized with Mullein extract, (**b**) UV-Vis spectra of stability of AuNPs incorporated in silica particles by synthesis-1, (**c**) UV-Vis spectra of stability of AuNPs incorporated in silica particles by synthesis-2.

**Figure 6 materials-15-07635-f006:**
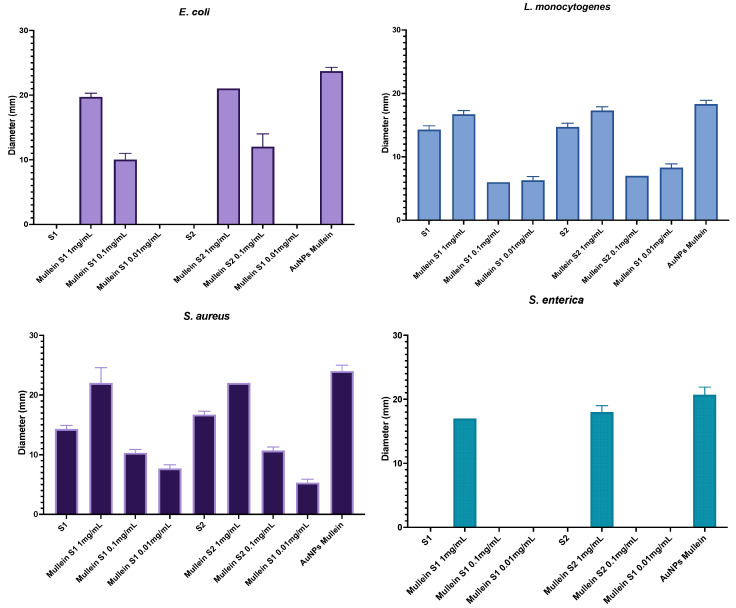
Antibacterial activity of silica-gold nanoparticles and AuNPs in different concentrations.

**Table 1 materials-15-07635-t001:** The diameter size and EDS composition of silica-gold nanoparticles.

Sample	Diameter (nm)	Si (%)	O (%)	Au (%)
Blank S1	196.96 ± 9.19	90.83 ± 0.62 ^a^	9.16 ± 0.60 ^a^	nd
Mullein S1-AuNPs	116.22 ± 2.04 *	89.16 ± 2.34 ^a^	8.53 ± 1.96 ^a^	2.33 ± 0.43 ^c^
Blank S2	560.32 ± 32.82	80.33 ± 0.88 ^a^	19.66 ± 0.88 ^a^	nd
Mullein S2-AuNPs	263.99 ± 18.65 *	70 ± 7.08 ^a^	20.16 ± 4.1 ^a^	7.73 ± 1.86 ^a^

Results are reported as means of three replicates ± standard error; means in each column bearing different superscripts (a and c) are significantly different (*p* < 0.01) according to the Tukey test (n = 3), only for diameter results, a Dunnet test was used to compare with control (Blank) difference are represented with *.

**Table 2 materials-15-07635-t002:** Antibacterial activity of silica–gold nanoparticles with different pathogenic microorganisms, inhibition zone in mm.

Sample	*E. coli*	*L. monocytogenes*	*S. aureus*	*S. enterica*
Blank S1	0.0 ± 0.0 *^d^	14.3 ± 0.6 ^b^	14.3 ± 0.6 ^b^	0.0 ± 0.0 *^b^
Mullein S1-AuNPs	19.7 ± 0.6 ^b^	16.7 ± 0.6 ^a^	22.0 ± 2.6 ^a^	17.0 ± 0.0 ^a^
Blank S2	0.0 ± 0.0 *^d^	14.7 ± 0.6 ^b^	16.7 ± 0.6 ^b^	0.0 ± 0.0 *^b^
Mullein S2-AuNPs	21.0 ± 0.0 ^a^	17.3 ± 0.6 ^a^	22.0 ± 0.0 ^a^	18.0 ± 1.0 ^a^
Mullein AuNPs	23.7 ± 0.6 ^a^	18.3 ± 0.6 ^a^	24.0 ± 1.0 ^a^	20.7 ± 1.2 ^c^

Results are reported as means of three replicates ± standard error; means in each column bearing different superscripts (a–d) are significantly different (*p* < 0.01) according to the Tukey test (n = 3). * The population decreased where the sample was placed, but a halo of inhibition was not formed. The concentration of particles for antibacterial activity reported is 1 mg/mL.

## Data Availability

Not applicable.

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
