# Peer review of "Comparative Study between Two Simple Synthesis Methods for Obtaining Green Gold Nanoparticles Decorating Silica Particles with Antibacterial Activity"

_materials, 2022, doi:10.3390/ma15217635_

Round 1

Reviewer 1 Report

In the submitted manuscript entitled 'Comparative study between two simple synthesis methods for obtaining green gold nanoparticles decorating silica particles with antimicrobial activity' the authors presented the synthesis and characterization of gold nanoparticles using mullein and castor extracts and its incorporation into silica particles in order to improve their colloidal stability while maintaining their properties. 

Despite the effort of the authors to present and discuss the experimental data obtained from the synthesis of these nanomaterials, I consider that the work does not represent any novelty in the field in terms of synthesis, characterization or material properties. The authors should improve the text to clearly introduce the objective of the work, properly describe the experimental data, discuss the effect of the plant extracts or even provide some hypotheses regarding the different antimicrobial activities of the gold-silica materials.

Specific comments:

1. The abstract should be rewritten in order to properly summarise the present work.

2. There are a lot of typos, e.g. the symbol of some units, the name of bacteria, and punctuation marks. I suggest to carefully revise the document before submission.

3. In the title and at the end of the introduction, the authors claimed that it represents a green synthesis since they are using mullein/castor extracts. What are the main components of these extracts? phenolic compounds? What about castor leaves extract? The role of these extracts should be studied and discussed, not only in the synthesis of the nanoparticles but also in the antimicrobial activity, since the phenolic compounds or other compounds in plant extracts are reported to have some antimicrobial properties.

4. Why is this synthesis greener than others previously reported in the literature?

5. From my point of view, the authors should present the synthesis of gold nanoparticles alone with mullein/castor extracts as control/reference.

6. What is the pH of the solution in synthesis 1 and 2? Are the plant extracts stable at these conditions?

7. The authors stated that 'one of the most comon ways to protect colloidal metal systems, like gold nanoparticles, from aggregation and loss of activity is their incorporation into other systems like SiO2 particles'. As can be observed in Figure 1, the materials resulting from syntheses 1 and 2 are clearly aggregated. From this image is not possible to determine the size or morphology of the particles.

Author Response

We greatly appreciate Reviewers #1 for their valuable comments and suggestions. You can find the answers to your comments in the attached file

Reviewer 2 Report

The manuscript entitled “Comparative study between two simple synthesis methods for obtaining green gold nanoparticles decorating silica particles with antimicrobial activity” could be interesting for readers. The manuscript is well-written on certain topics, structured, and technically sound. Following are the comments for improvement:

ABSTRACT

- Include type of samples tested (blank and mullein).

- Include important results from FTIR and XRD.

INTRODUCTION

- Literature on the two synthesis methods is not enough. Please add more literature.

- Following review must be included in the introduction:

Cong, V.T.; Ganbold, E.O.; Saha, J.K.; Jang, J.; Min, J.; Choo, J.; Kim, S.; Song, N.W.; Son, S.J.; Lee, S.B.; et al. Gold nanoparticle 282 silica nanopeapods. J. Am. Chem. Soc. 2014, 136, 3833–3841, doi:10.1021/ja411034q. 283

Yue, K.; Jin, X.; Tang, J.; Tian, X.; Tan, H.; Zhang, X. Factors Influencing Aggregation of Gold Nanoparticles in Whole Blood. J. 284 Nanosci. Nanotechnol. 2019, 19, 3762–3771, doi:10.1166/jnn.2019.16315.

Kim, K. Do; Kim, H.T. Formation of silica nanoparticles by hydrolysis of TEOS using a mixed semi-batch/batch method. J. Sol-273 Gel Sci. Technol. 2002, 25, 183–189, doi:10.1023/A:1020217105290. 274

Dabbaghian, M.; Babalou, A.; Hadi, P.; Jannatdoust, E. A Parametric Study of the Synthesis of Silica Nanoparticles via Sol-Gel 275 Precipitation Method. Int. J. Nanosci. Nanotechnol. 2010, 6, 104–113. 276

Rahman, I.A.; Padavettan, V. Synthesis of Silica nanoparticles by Sol-Gel: Size-dependent properties, surface modification, and 277 applications in silica-polymer nanocompositesa review. J. Nanomater. 2012, 2012, doi:10.1155/2012/132424.

MATERIALS AND METHODS

- Line 68 & 77: What is CTAB?

- Line 70: The dispersion is refer to what?

- Line 70: Under a vacuum. Please define the vacuum.

- Line 72: "...the blank S1[8]." Did the method 100% following the [8]? If yes, please mentioned in the paragraph.

- Line 83: Similarly to [9].

RESULTS AND DISCUSSION

- Please mention Figure 1(a),(b),(c),(d) in the paragraph. It is not enough to only mention Figure 1. 

- Figure 1(c) is not at similar magnification. Please add Figure 1(c) with 25,000x magnification. Comparison must be made at similar magnification.

- Please mention Figure 2(a),(b),(c),(d) in the paragraph. It is not enough to only mention Figure 2. 

- Figure 1(d) is not at similar magnification. Please add Figure 1(d) with 30,000x magnification.

- Line 145: Please mention the temperature changes/difference.

- Figure 3: Why Mullein S1 has a sharp peak at AU (200)?

- Figure 4: Why absorption bands at 1084 for Blank S2 shifted a bit as compare to Blank S1? And why the area is larger?

- Figure 4: Why FTIR results for Mullein S1 and S2 is quite similar? No distinct difference?

- Figure 4: Why all absorption bands for Blank S2 is shifted to the right as compared to Mullein S2?

- Line 182: It should be table 2.

CONCLUSION

- Acceptable.

Author Response

We greatly appreciate Reviewers #2 for their valuable comments and suggestions. You can find the answers to your comments in the attached file

Reviewer 3 Report

Synthesis of gold nanoparticles decorated by silica particles by green method and comparitive study between method of preparation by its physical properties are reported here. The novelty of the manuscirpt is quite low, the author should expose the novelty of the work clearly. This manuscript needs major revision before acceptance. 

1. The formation gold nanoparticles confirmed by XRD is not enough, however the absorption spectra must be studied

2. The stability of gold nanoparticles must be studied 

3. In both the synthesis, author used mullein extract only, so please delete the castor leaves from materials

4. The figures 1-4 must be resized

5. What is the MIC for bacterial pathogens you studied?

6. The names of bacterial species should be used unifrom and italic

7.  Are the morphologies of the two synthesized routes are similar with those references quoated for synthesis? 

Author Response

We greatly appreciate Reviewers #3 for their valuable comments and suggestions. You can find the answers to your comments in the attached file

Round 2

Reviewer 2 Report

Based on the response and revised manuscript, much effort has been given by the authors in correcting the manuscript. Most of the comments have been answered to the satisfaction of the reviewer. Although the quality and content of the manuscript can be further improved, this manuscript is suitable for publication. 

Author Response

We thank the reviewer for the comments received; we have improved the article in terms of grammar and typographical errors, in the same way, some corrections were made to the summary and data that are marked in the file.

Reviewer 3 Report

Authors addressed all comments. so I recommend the acceptance as is

Author Response

We thank the reviewer for his satisfactory response.